# Shortcut to chemically accurate quantum computing via density-based basis-set correction
Diata Traore[1,2], Olivier Adjoua[1], César Feniou[1,2], Ioanna-Maria Lygatsika[1,3,6,7], Yvon Maday[3,4], Evgeny Posenitskiy[2], Kerstin Hammernik [5], Alberto Peruzzo[2], Julien Toulouse [1,4], Emmanuel Giner[1] & Jean-Philip Piquemal [1,2] ✉

Using GPU-accelerated state-vector emulation, we propose to embed a quantum computing ansatz into density-functional theory via density-based basis-set corrections to obtain quantitative quantum-chemistry results on molecules that would otherwise require brute-force quantum calculations using hundreds of logical qubits. Indeed, accessing a quantitative description of chemical systems while minimizing quantum resources is an essential challenge given the limited qubit capabilities of current quantum processors. We provide a shortcut towards chemically accurate quantum computations by approaching the complete-basis-set limit through coupling the density-based basis-set corrections approach, applied to any given variational ansatz, to an on-the-fly crafting of basis sets specifically adapted to a given system and user-defined qubit budget. The resulting approach self-consistently accelerates the basis-set convergence, improving electronic densities, ground-state energies, and first-order properties (e.g. dipole moments), but can also serve as a classical, *a posteriori*, energy correction to quantum hardware calculations with expected applications in drug design and materials science.

Quantum computing (QC) offers a promising approach to solving electronic-structure problems, with algorithms like quantum phase estimation (QPE) and variational quantum eigensolver (VQE) proving effective in performing ground-state quantum-chemistry wave-function calculations[1–6]. The electronic Hamiltonian is expressed in second quantization, employing an encoding that maps one spin–orbital to one qubit. This mapping allows one to represent an exponentially large Hilbert space using only a linear number of qubits. However, to achieve accurate and practically valuable predictions for chemical systems in real-world applications, the molecular Hamiltonian should be expressed and solved using extensive basis sets of one-electron orbital functions. The number of qubits required for such calculations quickly exceeds the available capacities on current noisy intermediate scale quantum (NISQ) devices, upcoming early fault-tolerant quantum computing (FTQC) devices, and high-performance classical emulators. Therefore, while quantum chemistry has long been stated as a promising application for quantum computing, the endeavors have so far been limited to small molecular systems and minimal basis sets.

However, in practice, minimal basis sets fail to be predictive for ground-state energies, and chemically useful calculations require at least significantly larger than double-zeta basis sets. Requirements for computing molecular properties are even more drastic as they tend to converge more slowly with the size of the basis set than energies.

In electronic-structure theory, the exact solution is defined by the full-configuration interaction (FCI) method in the (infinite) complete-basis-set (CBS) limit. However, in practice, the solution derived from a finite basis set is used, which inherently suffers from truncation errors. These errors can be substantial for small basis sets, but for large enough basis sets it is possible to reach the target of chemical accuracy on energy differences, i.e., 1 kcal/mol (1.6 mHa). Unfortunately, employing sufficiently large basis sets becomes very rapidly impractical for large systems, particularly those with more than a few dozen atoms. As a result, quantum chemists have developed a variety of traditional (i.e., classical) computational methods to approach chemical accuracy at a reasonable cost[7–10]. Similarly, on the quantum computing side, the diversity of available methods has significantly increased[11–14].

[1]Sorbonne Université, LCT, UMR 7616 CNRS, 75005 Paris, France. [2]Qubit Pharmaceuticals, Advanced Research Department, 75014 Paris, France. [3]Sorbonne Université, LJLL, UMR 7598 CNRS, 75005 Paris, France. [4]Institut Universitaire de France, 75005 Paris, France. [5]NVIDIA Corporation, Santa Clara, CA, USA. [6]Present address: CEA, DAM, DIF, F-91297 Arpajon, France. [7]Present address: Université Paris-Saclay, LMCE, 91680 Bruyères-le-Châtel, France. ✉e-mail: jean-philip.piquemal@sorbonne-universite.fr

In particular, the hybrid quantum-classical VQE techniques initially designed to solve the general eigenvalue problem[5], have been shown to be particularly suited for chemical applications in the present NISQ era[5,15,16]. VQE algorithms have evolved over the years in two directions: (i) fixed-length ansätze, often inspired by classical coupled cluster, with approaches such as the unitary coupled cluster (UCC) and its extensions (see refs. 17–19 and references therein); (ii) adaptive methods such as the adaptive derivative-assembled pseudo-trotter ansatz variational quantum eigensolver (ADAPT-VQE)[20] have also started to be particularly popular since they allow tailoring system-specific ansätze with shorter circuits. In recent years, ADAPT-VQE approaches have been systematically improved[21–26] while alternative, resource-saving, adaptive techniques have also been introduced[27,28].

For applications to quantum computing, the inspiration coming from classical approaches has not been limited[14] to UCC-like techniques, and one can particularly mention the importance of explicitly correlated approaches[29,30]. Indeed, several quantum strategies originate from the latter and aim to maintain full accuracy while further minimizing quantum resources for obtaining more compact wave-function representations, which is the key criterion for hardware implementations. We can particularly mention the works on transcorrelated approaches[31–36], which introduce short-range correlation effects such as the electron–electron cusp condition[37] through various Hamiltonian modifications. Therefore, explicitly correlated quantum computing ansätze based on the transcorrelated approach has been shown to successfully evaluate ground-state energies while using fewer resources[38–42]. Some of them have even been extended to the evaluation of excited-state energies[39]. However, other strategies exist in classical quantum chemistry and could be particularly suited to quantum computing.

In particular, basis-set correction techniques such as the density-based basis-set correction (DBBSC) method[43–55], relying on density-functional theory (DFT), have proven to be effective for calculating ground- and excited-state energies, and also dipole moments, for a variety of systems, including atoms, small organic molecules, and strongly correlated systems. This approach provides the key benefit of an accelerated convergence to the CBS limit with the basis-set size, which is a valuable asset for quantum computing since minimizing quantum resources is paramount. To date, the DBBSC method has only been applied to still relatively large basis sets, more specifically the family of Dunning basis sets[56] for which chemical accuracy with respect to the CBS limit can be reached starting from a triple-zeta basis set. We note that the approximations developed within the DBBSC method target the CBS limit within a given wave-function ansatz. This means that they do not address the intrinsic errors of the wave-function method itself, such as the effects of neglected higher excitations in truncated configuration-interaction (CI) or coupled-cluster methods.

In this work, we propose the integration of the DBBSC method with quantum algorithms to expedite reaching the CBS limit and achieve chemical accuracy on complex molecular systems. This strategy, which natively limits the required qubit counts, can be applied to quantum algorithms that tackle the ground-state quantum chemistry problem, such as the QPE or VQE algorithms. While QPE can guarantee ground-state energy with arbitrarily high precision given a carefully chosen initial state, it demands large quantum circuits and will only be viable in the FTQC era. Conversely, VQE lacks convergence guarantees but employs smaller circuits, aligning better with this study's aim of advancing short-term quantum computers toward practical quantum-chemistry applications.

The paper is organized as follows. In Section "Methods", we present two variants of the application of the DBBSC method to wave-function QC calculations, denoted as Strategy 1 and Strategy 2: (1) a basis-set correction *a posteriori* added to the solution of the quantum algorithm, integrating two contributions, namely a basis-set correlation density-functional correction and a basis-set Hartree–Fock (HF) correction, and (2) a self-consistent scheme integrating the DBBSC method to the quantum algorithm that dynamically modifies the one-electron density used in the basis-set correlation correction. Strategy 1 offers the possibility to correct any wave-function QC energy calculation through a simple additive correction. Strategy 2 enables one to self-consistently access an improved electronic density, offering both improved energies and first-order molecular properties. Also, we introduce a new type of system-adapted basis sets (SABS) for Gaussian-type orbitals (GTOs), with sizes comparable to minimal basis sets. These new methodologies enable us to perform the first investigation of DBBSC corrections in the minimal basis-set regime.

In the Section "Results and discussion", we provide relevant tests of our approach for different atomic and molecular systems using several families of basis sets. We carry out numerical simulations using graphics-processing-unit (GPU) accelerated QC sparse emulation on up to 32 qubits, exploring the applicability of this method to converge ground-state energies, dissociation curves, and dipole moments. We consistently observe significant improvements over typical quantum algorithm approaches, reaching an accuracy level that would have otherwise required hundreds of qubits. We thus expect the present approach to become a standard part of quantum-enhanced wave-function calculations, particularly the approach of Strategy 1, which can immediately provide large improvements to existing results with relatively simple efforts.

Finally, the Section "Conclusion" contains our conclusions. Additional details and results are provided in the Supplementary Information (SI).

## Results and discussion

We computed the ground-state energies, dissociation curves, and dipole moments for the $N_2$, $H_2O$, LiH, and $H_2$ molecules using both basis-set correction strategies. For all systems except $H_2$, the 1s molecular orbitals were frozen, i.e., we use the frozen-core approximation. Correspondingly, we use the frozen-core version of the DBBSC method as defined in ref. 44, in which, in particular, the contribution to the density coming from the core electrons is neglected in the basis-set correlation density functional. The error in each calculation is quantified as the deviation from the CBS limit, which itself is determined using a two-point extrapolation scheme based on the cc-pVQZ and cc-pV5Z basis sets[57]. For the dipole–moment calculations we employed only Strategy 2, using an expectation value over the dipole–moment operator, avoiding a finite-difference approach prone to numerical errors[58]. We remind the reader that these calculations are to be interpreted in the framework of perfect (noiseless) logical qubits, and therefore the discussed errors can be only attributed to the method used. The Appendix presents the quantum circuits that were used for all computations and provides the number of CNOT gates required to perform for the algorithm as well as the number of ADAPT-VQE iterations.

The classical computations, including basis-set corrections and the calculation of reference energies and dipole moments, were carried out using Quantum Package 2.0[59]. In this software, the FCI energies are approximated by the energy from a CIPSI wave function to which a second-order perturbation theory (PT2) correction is added. Given the demonstrated nearly-FCI quality of this approximation for the systems we studied, we simplify our terminology by referring to this approach as simply FCI rather than CIPSI+PT2.

The SABS are labeled VXZ-Y where X is respectively D, T, Q, 5, or 6 for cc-pVDZ, cc-pVTZ, up to cc-pV6Z basis sets, and Y is the target size. We report in the SI the classically computed energies for the $N_2$, $H_2O$, LiH, and $H_2$ molecules.

### Ground-state energies

Ground-state energies of the $H_2$, LiH, $H_2O$, and $N_2$ molecules close to their respective equilibrium geometries are presented in Fig. 1 and Table 1. Details regarding the ADAPT-VQE iterations and the associated "ADAPT" values are provided in the SI, as well as additional tests on atoms and hydrogen chains.

In Fig. 1, a general trend is observed: the basis-set corrected ground-state energies with the small basis sets, i.e., STO-3G, 6-31G, or pcseg-0[60], align with values between the cc-pVDZ and cc-pVTZ basis-set levels, while requiring much fewer qubits than these latter basis sets. As we can see from Table 1, both strategies provide the same quantitative improvements. These

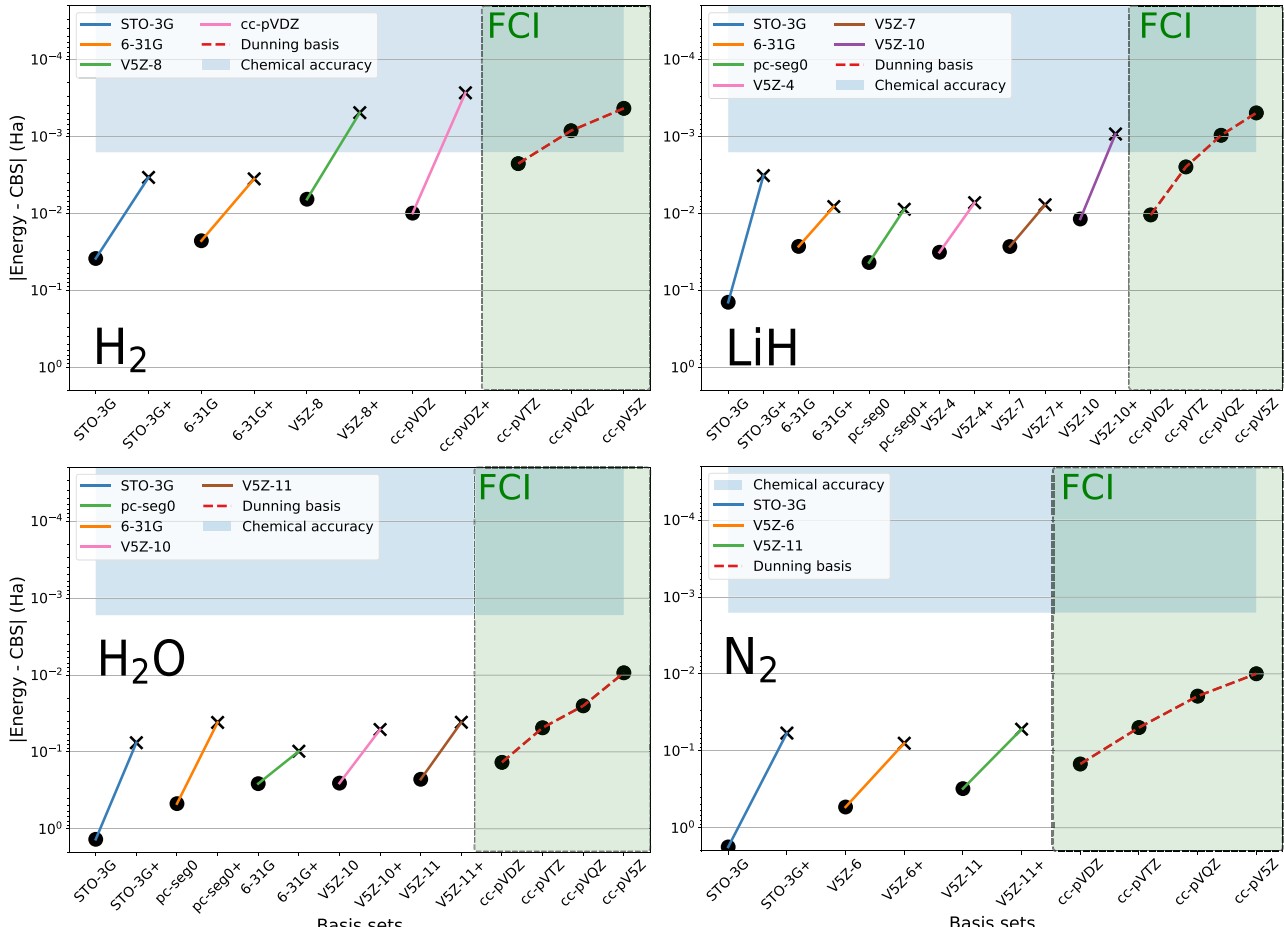

**Fig. 1 | Ground-state energy errors with respect to the extrapolated CBS limit.** Outside the green box: ADAPT-VQE calculations. Inside the green box: FCI calculations from a classical computer. Dot markers correspond to energies without basis-set corrections. Cross markers correspond to energies corrected with Strategy 1 [Eq. (6)]. On the x-axis, the labels $\mathcal{B}+$ symbolize basis-set corrected values. The blue box corresponds to a range of 1.6 mHa around the extrapolated CBS limit. The errors for each ADAPT-VQE computation with respect to the FCI energy for a given basis set are reported in the SI.

results are consistent with the conclusion from ref. 49 where the non-self-consistent approximation was found sufficient for calculating energies.

For the best cases, the basis-set corrected ground-state energies have errors in the order of tens of mHa: 40 mHa for $H_2O$ with 24 qubits, 60 mHa for $N_2$ with 16 qubits, and less than 10 mHa for LiH and $H_2$ with less than 10 qubits. The results stay consistent for the other systems available in the SI ($H_4$, $H_6$, $H_8$). For $H_2$ using the cc-pVDZ basis set, the basis-set corrected ground-state energy reaches chemical accuracy relative to the CBS limit while requiring 20 logical qubits. Similar convergence to the CBS limit is observed in the SI for He and Be. Without the basis-set correction, reaching a similar accuracy would have typically required more than a hundred qubits.

In the initial work on the DBBSC method[43], it was found that the basis-set corrected ground-state energies of atoms reach chemical accuracy with the cc-pVTZ basis set. Similarly, for the molecules studied here, if one had access to a few hundred logical qubits required for the cc-pVTZ basis set, achieving chemical accuracy would be theoretically feasible for all cases. To support this claim, we report in the SI classically computed basis-set corrected FCI ground-state energies using Dunning basis sets. The error relative to the CBS limit for the LiH molecule is 0.2 mHa with the cc-pVTZ basis set. For $H_2O$ and $N_2$, we achieve errors of 3 and 1 mHa, respectively, with the cc-pVTZ basis set.

Let us now discuss the results obtained with our SABS for LiH, $N_2$, and $H_2O$. They were chosen to match the sizes of the small basis sets previously discussed. Specifically for $H_2O$, employing the V5Z-10 basis set (24 qubits) achieves results slightly superior to those obtained with the 6-31G basis set.

For $H_2O$, we observed a reduction in the HF basis-set correction by 40 mHa when moving from the 6-31G to the V5Z-10 basis set, by over 100 mHa when comparing the STO-3G to the V5Z-4 basis set for the LiH molecule, and more than a Hartree for the $N_2$ molecule moving from the STO-3G to the V5Z-6 and V5Z-11 basis sets. Based on these findings, we anticipate that further exploration of this strategy could lead to a basis-set correction scheme requiring only the correlation basis-set correction term. Of course, the interest of SABS is to be systematically improvable toward the parent Dunning basis set. It is important to note that this SABS systematic improvement is not limited to quantum algorithms such as ADAPT-VQE and is also present for the HF and FCI calculations. Since the SABS approach strongly reduces the computational cost while maintaining accuracy, it should offer further applications of the CIPSI approach in classical quantum chemistry. In practice, a cc-pVTZ-like quality is reached with a V5Z-11 basis set (32 qubits) for $N_2$, and with a V5Z-11 basis set (30 qubits) for $H_2O$. A V5Z-10 basis set (28 qubits) achieves a cc-pV5Z-like ground-state energy accuracy for LiH. Overall, the SABS always provides the best ADAPT energies with the self-consistent correlation basis-set correction (before the addition of the HF basis-set correction).

Overall, concerning the basis sets, it is important to point out a key anomaly, i.e., the remarkable performance of the minimal STO-3G basis set. This was expected as Pople already discussed the outperformance of this basis set in the seventies[61]. Indeed, Davidson and Feller detailed in their 1986 review[62] the existence of error compensations and stated that "the smaller the basis the more ab initio calculations assume an empirical flavor". In practice, among the available minimal basis-set possibilities, STO-3G

**Table 1 | Ground-state energies (in Ha) for $H_2O$, $N_2$, LiH, $H_2$, and $H_8$ calculated by FCI, ADAPT-VQE (denoted as ADAPT), and basis-set corrected ADAPT-VQE according to Strategy 1, denoted as A + PBE + ΔHF, and to Strategy 2, denoted as SC(A + PBE) and SC(A + PBE) + ΔHF (i.e., without and with the HF basis-set correction, respectively)**

| $H_2O$ | $N_{qubits}$ | FCI | ADAPT | A + PBE + ΔHF | SC(A + PBE) | SC(A + PBE) + ΔHF |
|---|---|---|---|---|---|---|
| STO-3G | 12 | −75.01250 | −75.01250 | −76.30232 | −75.197880 | −76.30191 |
| pcseg-0 | 24 | −75.90855 | −75.90843[a] | −76.33681 | −76.03999 | −76.33279 |
| 6-31G | 24 | −76.11995 | −76.11989[b] | −76.28035 | −76.23717 | −76.32025 |
| V5Z-10 | 24 | −76.12626 | −76.12409 | −76.32705 | −76.27418[c] | −76.32505 |
| V5Z-11 | 30 | −76.15902 | −76.15165 | −76.33704 | −76.28622 | −76.33570 |
| cc-pVDZ | 46 | −76.24165 | – | – | – | – |
| cc-pVTZ | 114 | −76.33250 | – | – | – | – |
| cc-pVQZ | 228 | −76.35985 | – | – | – | – |
| cc-pV5Z | 400 | −76.36877 | – | – | – | – |
| CBS | – | −76.37812 | – | – | – | – |
| $N_2$ | $N_{qubits}$ | FCI | ADAPT | A + PBE + ΔHF | SC(A + PBE) | SC(A + PBE) + ΔHF |
| STO-3G | 16 | −107.65251 | −107.65251 | −109.36630 | −107.86974 | −109.36661 |
| V5Z-6 | 16 | −108.88869 | −108.88869 | −109.34552 | −109.09850 | −109.34608 |
| V5Z-11 | 32 | −108.89413 | −109.11566 | −109.37278 | −109.27385 | −109.37281 |
| cc-pVDZ | 52 | −109.27698 | – | – | – | – |
| cc-pVTZ | 116 | −109.37527 | – | – | – | – |
| cc-pVQZ | 216 | −109.40558 | – | – | – | – |
| cc-pV5Z | 360 | −109.41505 | – | – | – | – |
| CBS | – | −109.42498 | – | – | – | – |
| LiH | $N_{qubits}$ | FCI | ADAPT | A + PBE + ΔHF | SC(A + PBE) | SC(A + PBE) + ΔHF |
| STO-3G | 10 | −7.88218 | −7.88218 | −8.02160 | −7.89590 | −8.02119 |
| pcseg-0 | 14 | −7.98139 | −7.98139 | −8.0160 | −7.99166 | −8.01561 |
| 6-31G | 20 | −7.99800 | −7.99800 | −8.01668 | −8.00806 | −8.01611 |
| V5Z-4 | 10 | −7.99287 | −7.99287 | −8.01758 | −8.00562 | −8.01690 |
| V5Z-7 | 16 | −7.99793 | −7.99793 | −8.01710 | −8.01109 | −8.01643 |
| V5Z-10 | 28 | −8.01302 | −8.01302 | −8.02575 | −8.02134 | −8.02540 |
| cc-pVDZ | 26 | −8.01438 | – | – | – | – |
| cc-pVTZ | 86 | −8.02234 | – | – | – | – |
| cc-pVQZ | 190 | −8.02386 | – | – | – | – |
| cc-pV5Z | 290 | −8.02433 | – | — | – | – |
| CBS | – | −8.02482 | – | – | – | – |
| $H_2$ | $N_{qubits}$ | FCI | ADAPT | A + PBE + ΔHF | SC(A + PBE) | SC(A + PBE) + ΔHF |
| STO-3G | 4 | −1.13415 | −1.13415 | −1.17606 | −1.15590 | −1.17594 |
| 6-31G | 8 | −1.15003 | −1.15003 | −1.16911 | −1.16196 | −1.16913 |
| cc-pVDZ | 20 | −1.16275 | −1.16275 | −1.17239 | −1.16858 | −1.17246 |
| V5Z-8 | 24 | −1.16613 | −1.16613 | −1.17315 | −1.17170 | −1.17320 |
| cc-pVTZ | 56 | −1.17041 | – | – | – | – |
| cc-pVQZ | 120 | −1.17182 | – | – | – | – |
| cc-pV5Z | 220 | −1.17223 | – | – | – | – |
| CBS | – | −1.17265 | – | – | – | – |
| $H_8$ | $N_{qubits}$ | FCI | ADAPT | A + PBE + ΔHF | SC(A + PBE) | SC(A + PBE) + ΔHF |
| STO-3G | 16 | −4.24339 | −4.24320 | −4.45483 | −4.32764 | −4.45354 |
| 6-31G | 32 | −4.37032 | −4.35752 | −4.43488 | −4.41275 | −4.43502 |
| cc-pVDZ | 80 | −4.42756 | – | – | – | – |
| cc-pVTZ | 222 | −4.47121 | – | – | – | – |
| cc-pVQZ | 474 | −4.47702 | – | – | – | – |
| cc-pV5Z | 864 | – | – | – | – | – |
| CBS | – | – | – | – | – | – |

Here, PBE refers to the PBE-based correlation basis-set correction ΔHF refers to the HF basis-set correction, and SC stands for "self-consistent". The frozen-core approximation has been used for $H_2O$, $N_2$, and LiH. The CBS limits are estimated by two-point extrapolations from cc-pVQZ and cc-pV5Z calculations.
[a]500 iterations with a 14,668-determinant CIPSI initial state.
[b]1000 iterations with a 10,879-determinant CIPSI initial state.
[c]The value is −76.27636 when using an 11,016-determinant CIPSI initial state.

proved to be the most robust and cost-efficient choice for the basis-set corrections. Also, in the DBBSC schemes, the STO-3G basis set highly benefits from the HF basis-set correction, which reduces as the basis set and the number of qubits increases. When compared to SABS, for example, STO-3G always displays smaller self-consistent basis-set corrected ADAPT values and larger HF basis-set corrections. In practice, Pople also noted that STO-3G is especially good for energies around the equilibrium geometry, and larger basis sets are clearly required to describe accurately the notoriously more difficult dissociation curves and dipole moments.

Finally, one last aspect of the analysis of such large basis-set simulations is related to the convergence of ADAPT-VQE. Indeed, in VQE-type computations[5], there is no formal guarantee of convergence, and ADAPT-VQE belongs to such a heuristic family of methods. If, for systems like $H_2$, LiH, or $N_2$ (see SI, Figs. 3–14), convergence is obtained in a few dozen iterations, for more complex systems such as $H_2O$, the number of required iterations strongly increases when a double-zeta basis set is used. Clearly, Figure 2 in the SI shows that more than a thousand iterations are required to achieve full convergence starting from the HF initial state. This heuristic aspect means that no anticipation of the exact required number of iterations can be made. This may be manageable on a real quantum processor, but the use of classical emulation makes each iteration more costly in terms of time-to-solution than the previous one, limiting the overall convergence capabilities. It is possible to force convergence by replacing the HF starting point with a CIPSI one. Figure 2 in the SI shows that chemical accuracy can be easily reached with a well-converged CIPSI solution. However, ADAPT-VQE struggles to improve the solution, which is expected due to the enormous size of the parameter space. Furthermore, the CIPSI-based initial state already contains a significant amount of correlation. This represents a hard challenge for the ADAPT-VQE procedure, which now needs to pick the next ansatz operator to improve the existing quantum state. In any case, starting from an unconverged CIPSI wave function is a robust solution to strongly reduce the overall computational time. One example is given for $H_2O$ and the V5Z-10 basis set which initially led to a not fully converged result. With a better CIPSI starting point, it is possible to recover a few milli-Hartrees (see footnotes in Table 1). It is also possible to change the operator pool, but the point here is that convergence becomes challenging when tackling complex electronic structures, and such computations would not be possible without GPU-accelerated emulation.

### Dissociation curves

We pursue the dissociation energies reported in Fig. 2. Clearly, the simple ADAPT-VQE/STO-3G level of theory appears quite far from an accurate description of the dissociation (compared to the calculations in the largest basis sets), which is achieved with FCI/triple-zeta (and beyond) levels. The non-corrected ADAPT-VQE values are represented with bold lines, whereas the corrected values are reported with dashed curves. These dissociation curves are extremely challenging as they involve several regimes of correlation going from weak correlation happening typically at short distances to strong correlation effects at large distances. First, we notice that for all the cases, the basis-set corrected values around the equilibrium are always substantially closer to the large cc-pV5Z reference values than the uncorrected ones and improve with the basis-set size and the use of SABS. For larger distances, the basis-set requirements appear even more stringent. In practice, the three dissociation curves manage to converge to high accuracy thanks to the use of SABS: we obtain nearly a triple-zeta quality for $N_2$ using the basis-set correction with the V5Z-6 basis set (16 qubits). A triple-zeta-like quality can be achieved using the larger V5Z-11 basis-set at the price of more qubits (32 qubits, see Table 2). In the same line, nearly a cc-pV5Z quality for $H_2$ using the basis-set correction with the V5Z-8 basis set, and nearly a cc-pV5Z quality for LiH using the basis-set correction with a V5Z-7 basis set. For $H_2$, the basis-set corrected cc-pVDZ curve matches the cc-pV5Z reference up to a distance of 2.5 Å, and a slight discrepancy appears at long distances. This is consistent with classical computations leading to convergence to the CBS limit only when the basis-set correction is applied to the cc-pVTZ basis set (114 qubits). It is possible to fix this issue more

affordably by using a V5Z-8 basis set requiring only 24 qubits. For LiH, it is also possible to fix the small discrepancies observed on the V5Z-7 basis-set dissociation curve using a larger V5Z-10 basis set (28 qubits) which reaches the CBS limit.

Finally, it is important to highlight the good performance of the DBBSC method for the prediction of the dissociation curve of the triple-bonded $N_2$ molecule. Indeed, such computation is well-documented in the literature and known as extremely difficult as it requires both a multi-reference treatment and various weak correlation effects going from short-range to charge polarization effects[63–66]. At the cost of a minimal basis set (i.e., 16 qubits), our $N_2$ DBBSC computations achieved nearly a cc-pVTZ quality, providing an accuracy that would have required around 100 logical qubits in the context of a brute-force simulation. To the best of our knowledge, these results are the most accurate using a quantum algorithm when compared to the recent results from the literature. Indeed, a group of researchers managed to predict this dissociation curve using a Local Unitary Cluster Jastrow (LUCJ) ansatz coupled to double-zeta basis sets (6-31G and cc-pVDZ)[67]. The computation required the use of massive computational resources, namely hundreds of compute nodes of the Fugaku classical supercomputer coupled to a QPU. Alternatively, in the context of the fermionic quantum emulator, $N_2$ simulations used ADAPT-VQE coupled to the 6-31G and def2-SVP (56 qubits) basis sets to perform computation via approximate Matrix Product States (MPS)[68]. In the present work, we achieve a better accuracy with far fewer qubits and, again, these results can be systematically improved using larger SABS. Indeed, access to a cc-pVTZ-like basis set accuracy is possible using the next (larger) SABS in terms of size, i.e., V5Z-11 (32 qubits).

### Dipole moments

We also explore the idea of extending the basis-set correction scheme in QC calculations of molecular properties such as the dipole moment. As pointed out by Halkier et al.[69], the dipole moment also suffers from a slow basis-set convergence, which can be thought of as an indirect impact of the missing short-range correlation effects in a finite basis set. Therefore, it is relevant to apply the basis-set correction to the dipole moment, as already shown in ref. 49. We report in Table 2 the dipole moments of $H_2O$ and LiH. We calculate the basis-set corrected dipole moment as the expectation value over the dipole-moment operator over the last ADAPT-VQE wave function coming out of the self-consistent basis-set correction method. We also add *a posteriori* the HF basis-set correction to the dipole moment, calculated as the difference between the HF dipole moment in the aug-cc-pV5Z basis set and the HF dipole moment in the considered basis set. From Table 2, we see that the correlation basis-set correction is not sufficient to converge the dipole moments. However, the addition of the HF basis-set correction significantly improves the basis-set convergence. For both molecules, strong improvements are observed with respect to the ADAPT-VQE/STO-3G level. Further improvements are observed as the size of the basis set increases. Finally, we remark the ADAPT-VQE slow convergence in terms of iterations is also the source of small errors in the dipole moments.

### Conclusions

We demonstrated the applicability of the DBBSC method to QC algorithms for quantum chemistry. Using ADAPT-VQE, these approaches were shown to be able to systematically improve minimal basis-set results by predicting ground-state energies that are intermediate between double-zeta and triple-zeta FCI qualities. Overall, the presented self-consistent basis-set corrected ground-state energies are very close to their non-self-consistent counterparts. As the latter *a posteriori* basis-set correction approach can be very easily applied to any wave-function QC calculation performed on real quantum hardware, it provides an affordable improvement strategy for current QC chemistry computations. The self-consistent basis-set correction scheme is still useful since it permits the calculation of properties such as dipole moments thanks to the availability of improved QC densities.

An additional reduction of the required basis-set size is provided by our SABS approach. Besides being fast, as SABS can be generated within

**Fig. 2 | Dissociation curves of H₂, LiH, and N₂ molecules.** Blue and orange lines correspond to Dunning basis sets using the FCI method. The notation VXZ stands for the basis set cc-pVXZ. Green and purple plain lines correspond to ADAPT-VQE calculations without basis-set corrections. Green and purple dash-dotted lines correspond to basis-set corrected dissociation curves. Raw data are available in the SI.

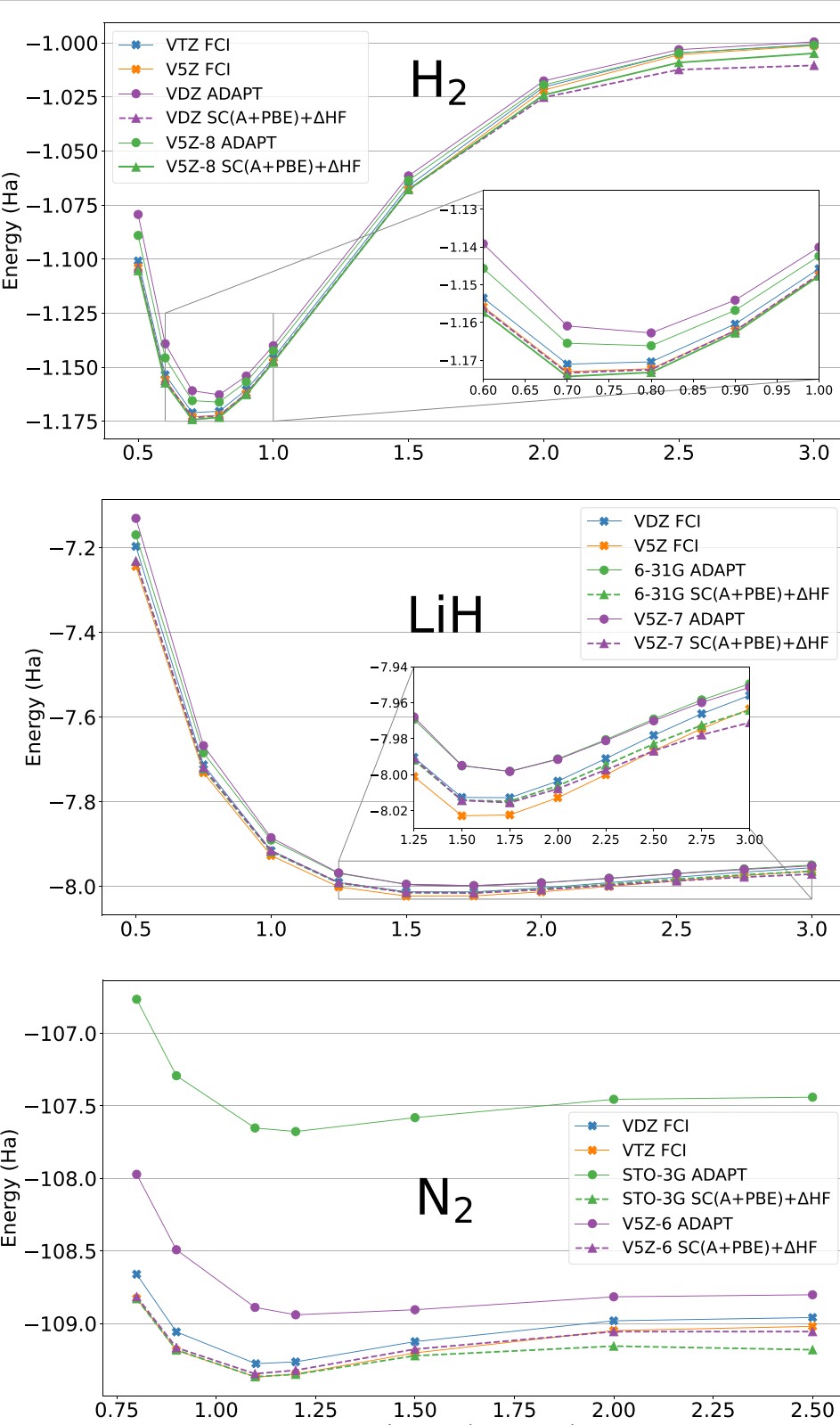

seconds, such a "black-box" pivoted Cholesky strategy for the on-the-fly generation of basis sets has been shown to be competitive and often superior to available standard choices. It offers systematic computational savings for large basis sets, reducing significantly the qubit requirements. Thus the SABS' usefulness is not limited to QC algorithms as they also offer systematically improvable solutions for performing reference classical computations. Indeed, since SABS are user-defined truncated versions of the Dunning basis sets that can match a qubit budget, they also offer reduced cost access to improved accuracy for any quantum-chemistry method.

Our methodology allows us to compute chemically meaningful energies and properties on systems that would have required far more than 100 logical qubits. For example, the computation of the H₂ total energy at the

**Table 2 | Dipole moments (in atomic units) of LiH and H₂O calculated by FCI and self-consistently basis-set corrected ADAPT-VQE, denoted as SC(A + PBE) and SC(A + PBE) + ΔHF (without and with and the HF basis-set correction, respectively)**

| LiH | $N_{qubits}$ | FCI | SC(A + PBE) | SC(A + PBE) + ΔHF |
|---|---|---|---|---|
| STO-3G | 10 | −1.81835 | −1.86299 | −2.31321 |
| pcseg-0 | 14 | −2.33313 | −2.37289 | −2.25650 |
| 6-31G | 20 | −2.16646 | −2.20768 | −2.23674 |
| V5Z-4 | 10 | −2.37818 | −2.44145 | −2.22886 |
| V5Z-7 | 16 | −2.23095 | −2.27789 | −2.25618 |
| V5Z-10 | 28 | −2.24997 | −2.27458 | −2.31438 |
| cc-pVDZ | 26 | −2.25566 | – | – |
| cc-pVTZ | 86 | −2.29998 | – | – |
| cc-pVQZ | 190 | −2.30361 | – | – |
| cc-pV5Z | 290 | −2.30647 | – | – |
| **H₂O** | $N_{qubits}$ | FCI | SC(A + PBE) | SC(A + PBE) + ΔHF |
| STO-3G | 12 | −0.63584 | −0.67084 | −0.77162 |
| pcseg-0 | 24 | −0.95822 | −0.99450 | −0.77065 |
| 6-31G | 24 | −0.99020 | −1.01898 | −0.76342 |
| V5Z-10 | 24 | −0.99305 | −1.01887 | −0.77634 |
| V5Z-11 | 30 | −0.99185 | −1.02170 | −0.77737 |
| cc-pVDZ | 46 | −0.76073 | – | – |
| cc-pVTZ | 114 | −0.75013 | – | – |
| cc-pVQZ | 228 | −0.74994 | – | – |
| cc-pV5Z | 400 | −0.74241 | – | – |

Here, ΔHF corresponds to the HF basis-set correction to the dipole moment, calculated as the difference between the HF dipole moment in the aug-cc-pV5Z basis set and the HF dipole moment in the considered basis set. More data are available in the SI.

FCI/cc-pV5Z level, which would have required more than 220 logical qubits, can be achieved here with only 24 qubits using our basis-set correction scheme and our SABS technique. Overall, we were able to converge four systems to the FCI/CBS limit, including He, Be, H₂, and LiH. We were also able to provide accurate dissociation curves for H₂, LiH, and N₂. Computations on H₂ and LiH required the use of only a single GPU. Since most quantum-chemistry studies are presently out of reach of quantum computers, this research opens the path to more affordable quantitative quantum-chemistry simulations of small molecules using QC algorithms. In particular, the present *a posteriori* basis-set corrections can be easily added to any type of STO-3G VQE fermionic computations on real hardware[5,70,71], allowing one to improve significantly their accuracy at very little computational cost. Since adaptive simulations on real hardware are making some progress[27] while the hardware itself improves, basis-set corrected simulations should be progressively possible on future quantum computers providing a route to FCI/CBS quality computations. This strategy is particularly suited for resources demanding computations that converge slowly and require large basis sets associated with large qubit counts. Finally, the DBBSC method is not limited to ground-state computations and can be extended to excited states via a linear-response formalism[53]. This will be the subject of future QC research.

In this context, besides the state-of-the-art ADAPT-VQE hybrid quantum-classical algorithm, it would be interesting to revisit accurate fixed wave-function ansätze[72], such as UCCSDT[73] and others, to analyze their convergence when coupled to the DBBSC method. In practice, the presented basis-set correction framework is not restricted to a given QC wave-function ansatz and can leverage any future QC algorithmic improvements. Our present DBBSC/SABS methodology still requires the use of qubits either through quantum hardware or through the use of a classical quantum

emulator. The qubit count is, therefore, our main limitation. The computations of this paper used up to 32 logical qubits and represent a proof-of-concept study of what one can presently do with quantum emulation to prepare the advent of fault-tolerant quantum computing[74], see Appendix for detailed resource estimations. However, state-vector simulations have limitations due to memory. Indeed, if they are theoretically possible up to 40-50 qubits on very large exascale supercomputers, they start to become relatively unpractical for quantum-chemistry simulations when reaching 36 qubits due to the high computational resources and time-to-solution requirements. To explore further the chemical electronic space with quantum algorithms, we are currently upgrading our Hyperion-1 framework to increase our emulated qubit counts by going beyond the state-vector formalism thanks to various elements from the CUDA-Q SDK[75,76] developed by NVIDIA. We are also presently testing various implementations of the DBBSC algorithms on available quantum hardware. To conclude, by reducing the number of qubits required to reach the CBS limit, we expect to tackle predictive real-world quantum chemistry applications with strategies applicable to both NISQ and FTQC algorithms.

## Methods

The overall methodological procedure is illustrated in Fig. 3. The procedure starts with the definition of the system, and a standard basis set or our SABS. After defining the second-quantized Hamiltonian, the quantum state is prepared on a quantum processing unit (QPU) or a GPU-accelerated quantum emulator. The basis-set corrections are calculated on the classical CPU using one of the two DBBSC variants labeled as Strategy 1 and Strategy 2.

### Density-based basis-set correction method

In the infinite-dimensional (antisymmetric) $N$-electron Hilbert space, $\mathcal{H} = \bigwedge^N L^2(\mathbb{R}^3 \times \{\uparrow, \downarrow\}, \mathbb{C})$, we consider an atomic or molecular system with Hamiltonian

$$\hat{H} = \hat{T} + \hat{W}_{ee} + \hat{V}_{ne}, \tag{1}$$

where $\hat{T}$ is the kinetic-energy operator, $\hat{W}_{ee}$ is the Coulomb electron-electron operator, and $\hat{V}_{ne}$ is the nuclei-electron potential operator. The exact ground-state energy is defined as

$$E_0 = \min_{\Psi \in \mathcal{W}} \langle \Psi, \hat{H}\Psi \rangle, \tag{2}$$

where $\mathcal{W} = \{\Psi \in \bigwedge^N H^1(\mathbb{R}^3 \times \{\uparrow, \downarrow\}, \mathbb{C}) \mid \langle \Psi, \Psi \rangle = 1\}$ is the space of admissible wave functions, $H^1$ is the first-order Sobolev space, and $\langle \cdot, \cdot \rangle$ designates the standard inner product of $\mathcal{H}$. In quantum-chemistry calculations, we normally introduce a one-electron basis set $\mathcal{B} \subset H^1(\mathbb{R}^3 \times \{\uparrow, \downarrow\}, \mathbb{C})$ and we work in the finite-dimensional $N$-electron Hilbert space generated by this basis set, i.e., $\mathcal{H}^{\mathcal{B}} = \bigwedge^N \text{span}(\mathcal{B})$. The FCI ground-state energy is then defined as

$$E_{FCI}^{\mathcal{B}} = \min_{\Psi \in \mathcal{W}^{\mathcal{B}}} \langle \Psi, \hat{H}\Psi \rangle, \tag{3}$$

where $\mathcal{W}^{\mathcal{B}} = \{\Psi \in \mathcal{H}^{\mathcal{B}} \mid \langle \Psi, \Psi \rangle = 1\}$ is the space of $\mathcal{B}$-representable wave functions. In the CBS limit, the FCI ground-state energy tends to the exact ground-state energy, i.e., $E_{FCI}^{\mathcal{B} \to CBS} = E_0$, but the convergence is infamously slow due to short-range electron correlation[57].

In the DBBSC method[43,49], for the given basis set $\mathcal{B}$, we introduce the following approximation to the ground-state energy

$$E_0^{\mathcal{B}} = \min_{\Psi \in \mathcal{W}^{\mathcal{B}}} \left( \langle \Psi, \hat{H}\Psi \rangle + \bar{E}^{\mathcal{B}}[n_\Psi] \right), \tag{4}$$

where $\bar{E}^{\mathcal{B}}[n_\Psi]$ is a basis-set correction density functional evaluated at the one-electron density of $\Psi$. This basis-set correction density functional must vanish in the CBS limit so that $E_0^{\mathcal{B}}$ properly converges to the exact ground-

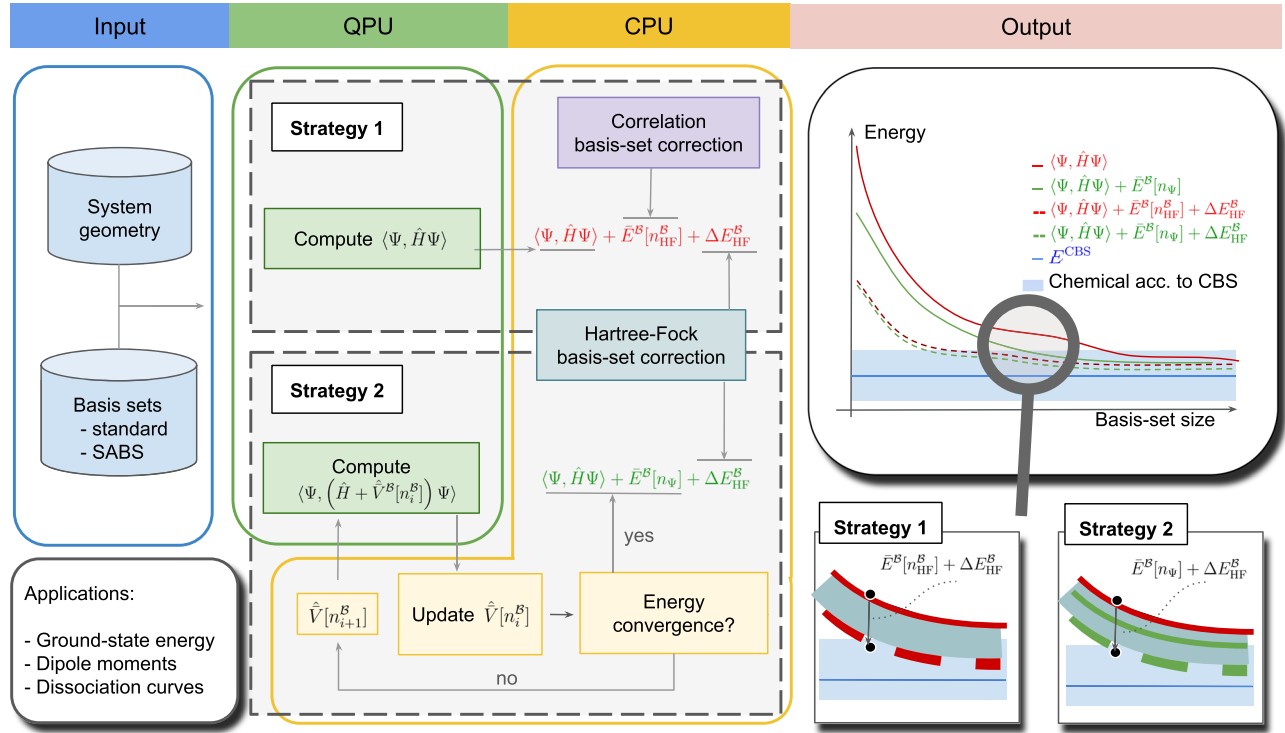

**Fig. 3 | Architecture of the hybrid quantum-classical scheme introducing the DBBSC method to wave-function QC calculations.** The QPU workload can be either performed by quantum hardware or replaced by a GPU-accelerated quantum emulator. The input are the definition of the system and the basis set (standard basis sets or our SABS can be used). In Strategy 1, the QPU/GPU computes the expectation value of the standard Hamiltonian $\langle\hat{H}\rangle$ and the non-self-consistent correlation basis-set correction and the HF basis-set correction are shifted to the CPU of the classical computer. In Strategy 2, the model includes a self-consistent basis-set correction potential which is iteratively optimized between QPU/GPU and CPU. The insets depict the effect of the basis-set corrections for the two strategies.

state energy, i.e., $E_0^{\mathcal{B}\to\text{CBS}} = E_0$, but it must be such that it accelerates the convergence to the CBS limit. In ref. 44, based on the Perdew–Burke–Ernzerhof (PBE) correlation density functional[77], such a basis-set correction density functional was constructed in a semilocal form

$$\bar{E}^{\mathcal{B}}[n] = \int_{\mathbb{R}^3} \bar{e}^{\mathcal{B}}(n(\mathbf{r}), \nabla n(\mathbf{r})) \, d\mathbf{r}, \quad (5)$$

where $n(\mathbf{r})$ and $\nabla n(\mathbf{r})$ are the density and density gradient at point $\mathbf{r}$, and the function $\bar{e}^{\mathcal{B}}(n, \nabla n)$, including how the dependence on the basis set $\mathcal{B}$ is included, can be found in ref. 44. We will now discuss how the DBBSC method can be adapted to wave-function QC calculations.

**Strategy 1: *a posteriori* basis-set correction.** In Strategy 1, we use a non-self-consistent approximation to Eq. (4). The idea is to calculate the FCI energy $E_{\text{FCI}}^{\mathcal{B}}$, or a good approximation to it, on a quantum computer, and then add *a posteriori* the basis-set correlation correction of Eq. (5) and a HF basis-set correction, both calculated on a classical computer. This leads to the following approximation to the ground-state energy

$$E_1^{\mathcal{B}} = E_{\text{FCI}}^{\mathcal{B}} + \bar{E}^{\mathcal{B}}[n_{\text{HF}}^{\mathcal{B}}] + \Delta E_{\text{HF}}^{\mathcal{B}}. \quad (6)$$

As in previous works[43,44], the basis-set correlation correction is evaluated at the HF density in the basis set $\mathcal{B}$, leading to a calculation of $\bar{E}^{\mathcal{B}}[n_{\text{HF}}^{\mathcal{B}}]$ with a marginal computational cost with respect to the cost of calculating $E_{\text{FCI}}^{\mathcal{B}}$. The PBE-based basis-set correlation correction only corrects for basis-set incompleteness errors due to short-range electron correlation. However, when using small basis sets, a significant part of the basis-set error also comes from the fact the HF part of the energy is not converged to the CBS limit. Similarly to Refs. 54,55, we thus add a HF basis-set correction. In the

present work, we choose it simply as the difference between the HF energy in the CBS limit, that we estimate as the value $E_{\text{HF}}^{5Z}$ obtained with the cc-pV5Z basis set[78], and the HF energy $E_{\text{HF}}^{\mathcal{B}}$ in the basis set $\mathcal{B}$

$$\Delta E_{\text{HF}}^{\mathcal{B}} = E_{\text{HF}}^{5Z} - E_{\text{HF}}^{\mathcal{B}}. \quad (7)$$

We note that it is also possible to avoid performing a HF calculation with the cc-pV5Z basis set for estimating the CBS limit of the HF energy by using a complementary auxiliary basis set, as in refs. 54,55, but we have not found it necessary for the present work.

Let us emphasize again that the PBE-based basis-set correction mentioned above only takes care of the correlation part and therefore does not correct the basis-set error of the HF energy. Hence, there is no double counting between the two basis-set corrections, as shown in Refs. 54,55.

**Strategy 2: self-consistent basis-set correction.** In Strategy 2, we use the self-consistent version of the DBBSC method in Eq. (4), using the basis-set correlation correction of Eq. (5) in the self-consistent part of the calculation, and we only add *a posteriori* the fixed HF basis-set correction. This leads to the following approximation to the ground-state energy that can be applied to any variational quantum ansatz

$$E_2^{\mathcal{B}} = \min_{\Psi\in\mathcal{W}^{\mathcal{B}}}\left(\langle\Psi,\hat{H}\Psi\rangle + \bar{E}^{\mathcal{B}}[n_\Psi]\right) + \Delta E_{\text{HF}}^{\mathcal{B}}. \quad (8)$$

The minimization in Eq. (8) leads to the following self-consistent Schrödinger equation[49]

$$\hat{P}^{\mathcal{B}}\hat{\bar{H}}^{\mathcal{B}}[n_{\Psi^{\mathcal{B}}}]\Psi^{\mathcal{B}} = \mathcal{E}^{\mathcal{B}}\Psi^{\mathcal{B}}, \quad (9)$$

**Table 3 | Walltime (in minutes) required to grow the wave-function ansatz of size $N_{adapt}$ for a given molecular system using Hyperion-1 state-vector emulator on $N_{gpus}$ NVIDIA GPUs**

| Molecule/basis set | $N_{adapt}$ | $N_{qubits}$ | $N_{gpus}$ | A100 walltime [min] | H100 walltime [min] |
|---|---|---|---|---|---|
| $H_2O$/6-31G | 500 | 24 | 1 | 644 | 503 |
| $H_{12}$/STO-3G | 500 | 24 | 1 | 174 | 134 |
| $H_{14}$/STO-3G | 300 | 28 | 8 | 283 | 184 |
| $H_{16}$/STO-3G | 100 | 32 | 128 | 450 | 147 |

The results have been obtained on A100 (80 GB) and H100 (80 GB) GPUs using CUDA Toolkit 12.0 and NVIDIA HPC SDK 23.3.

where $\hat{P}^{\mathcal{B}}$ is the projector on the $N$-electron Hilbert space generated by the basis set $\mathcal{B}$, i.e., $\mathcal{H}^{\mathcal{B}}$, and $\hat{\tilde{H}}^{\mathcal{B}}[n]$ is the effective Hamiltonian

$$\hat{\tilde{H}}^{\mathcal{B}}[n] = \hat{H} + \hat{\tilde{V}}^{\mathcal{B}}[n]. \qquad (10)$$

Here, $\hat{\tilde{V}}^{\mathcal{B}}[n]$ is the basis-set correction one-electron potential operator

$$\hat{\tilde{V}}^{\mathcal{B}}[n] = \int_{\mathbb{R}^3} \bar{v}^{\mathcal{B}}[n](\mathbf{r})\,\hat{n}(\mathbf{r})\,\mathrm{d}\mathbf{r}, \qquad (11)$$

where $\bar{v}^{\mathcal{B}}[n](\mathbf{r}) = \delta\bar{E}^{\mathcal{B}}[n]/\delta n(\mathbf{r})$ is the derivative of the basis-set correlation correction with respect to the density, and $\hat{n}(\mathbf{r})$ is the density operator.

The idea is now to solve iteratively Eq. (9) on a quantum computer. The potential $\bar{v}^{\mathcal{B}}[n](\mathbf{r})$, and therefore the Hamiltonian $\hat{\tilde{H}}^{\mathcal{B}}[n]$, is iteratively updated with the density $n_i^{\mathcal{B}}$ of the wave-function ansatz solution $\Psi_i^{\mathcal{B}}$ of the $i^{th}$ iteration. The convergence criterion is reached when the difference between the last two iterated energy eigenvalues $\mathcal{E}_i^{\mathcal{B}}$ is less than 0.1 mHa. In practice, this is fulfilled after two iterations.

Finally, the self-consistent basis-set corrected energy $E_2^{\mathcal{B}}$ can be directly computed from the last energy eigenvalue $\mathcal{E}^{\mathcal{B}}$ and density $n^{\mathcal{B}} = n_{\Psi^{\mathcal{B}}}$ coming out from the quantum solver as follows

$$E_2^{\mathcal{B}} = \mathcal{E}^{\mathcal{B}} + \bar{E}^{\mathcal{B}}[n^{\mathcal{B}}] - \int_{\mathbb{R}^3} \bar{v}^{\mathcal{B}}[n^{\mathcal{B}}](\mathbf{r})n^{\mathcal{B}}(\mathbf{r})\,\mathrm{d}\mathbf{r} + \Delta E_{HF}^{\mathcal{B}}, \qquad (12)$$

where the HF basis-set correction $\Delta E_{HF}^{\mathcal{B}}$ is identical to the one used in Strategy 1.

In Strategy 2, the density of the QC wave-function ansatz $\Psi$ needs to be calculated. For this, we calculate the one-particle density matrix

$$n_{pq,\sigma} = \langle\Psi, \hat{a}_{p,\sigma}^{\dagger}\hat{a}_{q,\sigma}\Psi\rangle, \qquad (13)$$

where $\hat{a}_{p,\sigma}^{\dagger}$ and $\hat{a}_{q,\sigma}$ are the creation and annihilation operators, respectively, for the $p$th and $q$th orbitals and with spin $\sigma \in \{\uparrow, \downarrow\}$. This is achieved with a Jordan-Wigner mapping of the operator $\hat{a}_{p,\sigma}^{\dagger}\hat{a}_{q,\sigma}$ and the state $\Psi$ prepared using the parameterized ansatz. The density matrix is then used to classically update the Hamiltonian in Eq. (10) and to calculate dipole moments.

**GPU-accelerated QPU emulation and shift of the basis-set correction to classical resources.** The DBBSC method does not increase the qubit count as the basis-set correction is performed classically using a density functional and a HF correction. In practice, such computations present no advantage to be performed at the QC level since they would consume a large number of qubits without accuracy benefit over its classical counterpart. Therefore a hybrid QC wave-function/classical DBBSC approach is highly preferable. In this work, VQE/DBBSC computations are performed classically on GPU-accelerated computing nodes, the GPUs replacing the quantum processing unit (QPU) through the use of a classical quantum emulator. The basis-set correction contributions can be then shifted to the unused CPUs to maximize the computational efficiency while the computationally challenging VQE

algorithm is fully offloaded on GPUs for optimal time-to-solution performances. Of course, the proposed hybrid schemes are also fully tractable on a real quantum computer, as the basis-set correction contributions can be entirely shifted to classical resources harnessing GPUs/CPUs to preserve the qubit count while the core wave-function ansatz is maintained on the QPU.

**System-adapted basis-set generation**
Within the DBBSC method, the natural choice for basis sets is the Dunning correlation-consistent basis-set family that offers a path to a reliable convergence to the CBS limit, as demonstrated in initial DBBSC classical quantum-chemistry studies[43–55]. For QC calculations, such basis sets are unfortunately usually out of reach of quantum hardware/emulators as they are associated with very large qubit counts. In the present section, we address the task of generating atomic-orbital (AO) basis sets under a basis-size budget, with controllable accuracy[79]. We propose a mathematical formulation of this goal in terms of a constraint optimization problem, as follows.

Given a molecule composed of $N_{atm}$ atoms, with fixed nuclear coordinates $\{\mathbf{r}_a\}_{1 \le a \le N_{atm}}$, let

$$\mathcal{B} = \{\chi_{\mu}\}_{1 \le \mu \le N_{bas}} = \bigcup_{a=1}^{N_{atm}} \{\chi_{\mu}^a(\mathbf{r} - \mathbf{r}_a)\}_{1 \le \mu \le N_{bas}^a}$$

denote a spatial basis set of atom-centered GTOs. It is assumed that a one-electron density, denoted by $n_0^{\mathcal{B}}$, is available after a converged ground-state energy minimization procedure (e.g., HF or CI). Our main purpose is to *a posteriori* extract subsets of the given "large" basis set $\mathcal{B}$, denoted by $\mathcal{B}_I = \{\chi_{\mu}\}_{\mu \in I} \subseteq \mathcal{B}$ for any index subset $I \subseteq \{1, \ldots, N_{bas}\}$, achieving (i) a target size, and (ii) minimal accuracy loss on the given density $n_0^{\mathcal{B}}$ used as a reference. In the present computing scheme, since the DBBSC method uses the cc-pV5Z basis set for the HF basis-set correction, we choose $\mathcal{B} = $ cc-pV5Z. In practice, this choice is motivated by the fact that such a quintuple-zeta basis set fairly approaches chemical accuracy compared to quasi-exact all-electron numeric atom-centered orbital computations[80]. Hence, given a target basis size $M$ smaller than $N_{bas}$, we seek the optimal index subset, denoted by $I_M$, that minimizes the *best approximation* error of the density $n_0^{\mathcal{B}}$ over the set of $\mathcal{B}_I$-representable one-electron densities, for any $I \subseteq \{1, \ldots, N_{bas}\}$ with $|I| = M$, where $|\cdot|$ denotes the cardinal of a set, or, in other words, we solve an optimization problem under constraints:

$$I_M := \arg\min_{\substack{I \subseteq \{1, \ldots, N_{bas}\} \\ |I| = M}} \min_{n^{\mathcal{B}_I}} \|n_0^{\mathcal{B}} - n^{\mathcal{B}_I}\|, \qquad (14)$$

where $\|\cdot\|$ is a given norm for functions over $\mathbb{R}^3$. In the present work, we use the norm induced by the Coulomb inner product $\langle u, v\rangle_c := \int_{\mathbb{R}^3}\int_{\mathbb{R}^3}\mathrm{d}\mathbf{r}\,\mathrm{d}\mathbf{r}'\,u(\mathbf{r})v(\mathbf{r}')|\mathbf{r} - \mathbf{r}'|^{-1}$. Let us emphasize that Eq. (14) requires knowledge of the density $n_0^{\mathcal{B}}$, which is assumed to be precomputed on a classical computer. Such quantity is available in the framework of the DBBSC scheme since HF computations are systematically performed at the cc-pV5Z level to estimate the CBS limit of the HF energy.

The problem in Eq. (14) can be solved as follows. Our approach is to identify a greedy procedure for discarding elements of the full AO-product set that spans the space containing the reference density $n_0^{\mathcal{B}}$, which admits the expansion

$$n_0^{\mathcal{B}} = \sum_{\mu,\nu=1}^{N_{\text{bas}}} D_{\mu\nu}\chi_\mu\chi_\nu,$$

where $D = CC^{\mathsf{T}}$ is the density matrix and $C$ is the $N_{\text{bas}} \times N_{\text{occ}}$ matrix of the coefficients in the AO basis of the $N_{\text{occ}}$ occupied molecular orbitals. We plan to achieve this by eliminating linear dependencies present in the AO-product set[81]. The pivoted Cholesky decomposition (PCD) of a matrix[82] is an algebraic tool for eliminating linear dependencies occurring between matrix rows (resp. columns), which may be interpreted as an iterative greedy procedure for discarding elements that do not contribute to the full row (resp. column) space, up to an orthogonal projection error tolerance. PCD has been previously applied to the auxiliary basis-set generation for density fitting[81,83,84]. In the present work, we employ PCD within a new scheme, named system-adapted basis-set (SABS) generation, for solving the problem in Eq. (14).

Let us formulate our scheme in detail. Prior to the AO-product selection and in order to ensure orbital symmetry of the resulting SABS, we pre-process the initial basis $\mathcal{B}$ and first contract all angular components (e.g., all three p-type components $p_x, p_y, p_z$) of GTOs. To this end, we consider the partition $\{B_i\}_{i=1}^{N_{\text{orb}}}$ of $\{1, \ldots, N_{\text{bas}}\}$, each $B_i$ being an index block containing all angular components of a single GTO in $\mathcal{B}$, and the $N_{\text{orb}} \times N_{\text{bas}}$ contraction matrix $P$, defined for any $1 \le i \le N_{\text{orb}}$ as $P_{ij} = 1$ if $j \in B_i$ and zero otherwise. Next, we define the four-index tensor $T$ with entries

$$T_{pqrs} = \sum_{\mu,\nu,\kappa,\lambda=1}^{N_{\text{bas}}} P_{p\mu} P_{q\nu} D_{\mu\nu} \langle \chi_\mu\chi_\nu, \chi_\kappa\chi_\lambda \rangle_c D_{\kappa\lambda} P_{r\kappa} P_{s\lambda},$$

and fold pairwise its first two and its last two dimensions, in order to form the $N_{\text{prod}} \times N_{\text{prod}}$ Gram matrix of weighted AO products, denoted by $G$, with $N_{\text{prod}} = (N_{\text{orb}})^2$. As a last pre-processing step, we discard rows and columns of $G$ corresponding to products made of components not centered on the same atom and denote the resulting submatrix $A$. Now, PCD is applied to $A$, using the machine epsilon as a tolerance threshold, yielding an index set sorted in pivot-descending order. Given a target $M$, we define the selected AO-product index set, denoted by $S_M^{\text{Chol}}$, as the $M$-first pivot indices. Lastly, we recover the underlying AO index set

$$J_M = \bigcup_{(p_1,p_2) \in S_M^{\text{Chol}}} \{p_1, p_2\},$$

and post-process it in order to ensure that the same GTO-parameter set is assigned to all atoms of the same chemical type. For this purpose, the new basis set associated to a chemical type is the union of parameters of those GTOs in $\mathcal{B}_{J_M}$ that are centered on any atom of that type. This yields a solution $I_M \supseteq J_M$ to our problem in Eq. (14) and the resulting SABS is $\mathcal{B}_{I_M}$.

Note that our generation scheme directly fixes the size $M$ of the selected products, i.e., $|S_M^{\text{Chol}}| = M$. The actual size of the AO basis set, equal to $|I_M|$, is only implicitly controlled during our procedure. In practice, as numerical results show, $|I_M|$ is very close to $M$ for s- and p-type basis sets, while it remains the same order as $M$ for higher angular-momentum orbital types. Overall, the SABS generation approach is extremely fast and offers access to compact basis sets, specifically adapted to a given system and user-defined qubit budget. Examples of SABS generation can be found in the SI.

We note that the adaptation and generation of basis sets to the molecular geometry has been the subject of several publications exploring other strategies. Among them, we can cite some recent works[85,86], but also some other research in the context of quantum computing focusing on the need to

**Table 4 | CNOT-gate counts for qubit and qubit-excitation-based (QEB) operator pools**

| $H_2$ | $N_{\text{qubits}}$ | $N_{\text{CNOT}}$ (if QEB) | $N_{\text{CNOT}}$ (if qubit) | $N_{\text{op}}$ | $N_{\text{adapt iter}}$ |
|---|---|---|---|---|---|
| STO-3G | 4 | 13 | 6 | 1 | 1 |
| 6-31G | 8 | 71 | 34 | 7 | 7 |
| cc-pVDZ | 20 | 233 | 110 | 21 | 21 |
| V5Z-8 | 24 | 395 | 186 | 35 | 35 |
| $H_4$ | $N_{\text{qubits}}$ | $N_{\text{CNOT}}$ (if QEB) | $N_{\text{CNOT}}$ (if qubit) | $N_{\text{op}}$ | $N_{\text{adapt iter}}$ |
| STO-3G | 8 | 207 | 98 | 19 | 19 |
| $H_6$ | $N_{\text{qubits}}$ | $N_{\text{CNOT}}$ (if QEB) | $N_{\text{CNOT}}$ (if qubit) | $N_{\text{op}}$ | $N_{\text{adapt iter}}$ |
| STO-3G | 12 | 2437 | 1134 | 199 | 199 |
| $H_8$ | $N_{\text{qubits}}$ | $N_{\text{CNOT}}$ (if QEB) | $N_{\text{CNOT}}$ (if qubit) | $N_{\text{op}}$ | $N_{\text{adapt iter}}$ |
| STO-3G | 16 | 12677 | 5870 | 999 | 999 |
| 6-31G | 32 | 8855 | 4090 | 685 | 685 |
| He | $N_{\text{qubits}}$ | $N_{\text{CNOT}}$ (if QEB) | $N_{\text{CNOT}}$ (if qubit) | $N_{\text{op}}$ | $N_{\text{adapt iter}}$ |
| pc-seg0 | 4 | 19 | 10 | 3 | 3 |
| 6-31G | 4 | 19 | 10 | 3 | 3 |
| cc-pVDZ | 10 | 58 | 28 | 6 | 6 |
| Be | $N_{\text{qubits}}$ | $N_{\text{CNOT}}$ (if QEB) | $N_{\text{CNOT}}$ (if qubit) | $N_{\text{op}}$ | $N_{\text{adapt iter}}$ |
| STO-3G | 8 | 39 | 18 | 3 | 3 |
| pc-seg0 | 10 | 58 | 28 | 6 | 6 |
| 6-31G | 16 | 207 | 98 | 19 | 19 |
| cc-pVDZ | 26 | 292 | 136 | 26 | 26 |
| LiH | $N_{\text{qubits}}$ | $N_{\text{CNOT}}$ (if QEB) | $N_{\text{CNOT}}$ (if qubit) | $N_{\text{op}}$ | $N_{\text{adapt iter}}$ |
| STO-3G | 10 | 90 | 44 | 10 | 10 |
| pc-seg0 | 14 | 258 | 124 | 26 | 26 |
| 6-31G | 20 | 511 | 242 | 47 | 47 |
| VQZ-4 | 10 | 90 | 44 | 10 | 10 |
| V5Z-4 | 10 | 90 | 44 | 10 | 10 |
| V5Z-7 | 16 | 291 | 138 | 27 | 27 |
| V5Z-10 | 28 | 1083 | 506 | 91 | 91 |
| $H_2O$ | $N_{\text{qubits}}$ | $N_{\text{CNOT}}$ (if QEB) | $N_{\text{CNOT}}$ (if qubit) | $N_{\text{op}}$ | $N_{\text{adapt iter}}$ |
| STO-3G | 12 | 729 | 342 | 63 | 63 |
| 6-31G | 24 | 12657 | 5862 | 999 | 999 |
| V5Z-11 | 30 | 12647 | 5858 | 999 | 999 |
| $N_2$ | $N_{\text{qubits}}$ | $N_{\text{CNOT}}$ (if QEB) | $N_{\text{CNOT}}$ (if qubit) | $N_{\text{op}}$ | $N_{\text{adapt iter}}$ |
| STO-3G | 16 | 4868 | 2256 | 386 | 386 |
| V5Z-6 | 16 | 9714 | 4492 | 758 | 758 |
| V5Z-11 | 32 | 11718 | 5420 | 916 | 916 |

$N_{\text{op}}$ is the number of operators in the final wave-function ansatz, and $N_{\text{adapt iter}}$ is the number of ADAPT-VQE iterations achieved and at which we collect the values in the Table.
The numbers of CNOT gates, $N_{\text{CNOT}}$, are evaluated as $N_{\text{single}}^3 + 13 N_{\text{double}}$ for the QEB pool, and as $N_{\text{single}}^2 + 6 N_{\text{double}}$ for the qubit pool, where $N_{\text{simple}}$ is the number of single-qubit operators and $N_{\text{double}}$ is the number of two-qubit operators.

limit the qubit requirements through basis-set reoptimization such as refs. 87,88. Alternatively, Kottman and Aspuru[89] proposed a basis-set-free approach through an adaptive representation using pair-natural orbitals which was tested up to 22 qubits.

## Computational details

In the present study, we perform ADAPT-VQE computations[20] using the Qubit-Excitation-Based pool of operators, which is considered a standard[22]. Additional details about the ADAPT-VQE methodology can be found in the SI. The communication of the Hamiltonian from the CPU to the QPU/GPU is done by using standard FCIDUMP files to communicate the one- and two-electron integrals to the QPU/GPU software in order to construct the fermion operators. TREXIO files[90] are also useful to communicate a wider range of relevant information.

ADAPT-VQE computations were performed using the Hyperion-1 GPU-accelerated state-vector sparse emulator[91] up to 32 qubits. Hyperion-1 uses classical computing systems and is grounded on an efficient multi-GPU ensemble of fast custom sparse linear-algebra libraries accelerating Hyperion-1's exact/noiseless simulations. In this paper, computations were performed on NVIDIA DGX A100 nodes (8× 80GB A100 GPUs per node) and NVIDIA DGX H100 nodes (4× & 8× 80 GB H100 GPUs per node). QC calculations being strongly memory-dependent, a single GPU can carry out a 20-qubit ADAPT-VQE simulation depending on the nature (i.e., Hamiltonian sparsity) of the system whereas a single node (8 GPUs) can handle up to 28 qubits. Multi-node computations are required beyond such a qubit count. Further details about Hyperion-1 and its full capabilities will be given in a forthcoming publication. Convergence for all ADAPT-VQE computations were set to $10^{-6}$ Ha. Most computations started from a HF initial state. For selected ones (indicated in the text and Tables), we started the ADAPT-VQE procedure from a rough configuration-interaction perturbatively-selected-iteratively (CIPSI)[92] initial state (converged to only $10^{-2}$ Ha) to save some computational time within Hyperion-1. This reflects a commonly adopted strategy where a multi-determinant initial state is employed instead of a single HF determinant to increase the ground-state support in the initial state. The *Quantum State Preparation* of such classically-derived CIPSI wave functions has been studied in the context of VQE[25] and QPE[26]. Also, we report in Table 3, the walltime required for several ground-state energy calculations. As one can see, the walltime is not only a function of the number of qubits, sparsity is also important. For example, the Hamiltonian of the $H_{12}$ molecule is sparser than the one of the water molecule resulting in a faster convergence. Besides the increased computing power of H100 GPUs leading to improved time-to-solution, our results also highlight the importance of fast node-to-node interconnects when performing large-scale quantum emulation. Indeed, the benefit of H100 over A100 is striking for the largest 32 qubits simulations on 16 nodes where an improvement of factor 3 was observed on the DGX H100 systems. Such speedup is, therefore, also partially related to higher node-to-node bandwidth observed on DGX H100 versus A100 systems.

## Appendix: CNOT counts

CNOT-gate counts for qubit and qubit-excitation-based (QEB) operator pools are presented in Table 4.

## Data availability

Data generated during the study is available upon request from the authors (E-mail: jean-philip.piquemal@sorbonne-universite.fr).

## Code availability

The code used during the study is available upon request from the authors (E-mail: jean-philip.piquemal@sorbonne-universite.fr).

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

## Acknowledgements

We thank the PEPR EPIQ—Quantum Software (ANR-22-PETQ-0007, J.-P.P.) for funding D. Traore's initial postdoctoral position. Support of the HQI initiative is also acknowledged (J.-P.P. and Y.M.). I.-M. Lygatsika's PhD position has been funded by the European Research Council (ERC) under the European Union's Horizon 2020 research and innovation program (grant No. 810367, project EMC2, J.-P.P. and Y.M.). Computations have been performed at IDRIS (Jean Zay) on GENCI Grants: No. A0150712052 (J.-P.P.) and grant GC010815453 (Grand Challenge H100 Jean Zay, J.-P.P.), at Scaleway (Jero) and on the SuperPOD reference cluster EOS at NVIDIA. The authors thank D. Ruau and C. Hundt (NVIDIA) for continuous support, as well as the NVIDIA Quantum team for technical discussions.

## Author contributions

D.T., O.A., I.M.L., E.P., and K.H. performed the computations. O.A., D.T., C.F., and E.P. wrote the quantum emulation code. D.T., O.A., I.M.L., Y.M., E.G., A.P., J.T., E.G., and J.P.P. analyzed the data. D.T., O.A., C.F., I.M.L., Y.M., E.P., J.T., E.G., and J.P.P. contributed to the methodology. D.T., O.A., I.M.L., A.P., J.T., E.G., and J.P.P. wrote the paper with inputs from all other co-authors. J.P.P. designed and supervised the research.

## Competing interests

J.P.P. is a shareholder and co-founder of Qubit Pharmaceuticals. The remaining authors declare no other competing interests.
