## [Transparent Peer Review file · Communications Chemistry]

Shortcut to Chemically Accurate Quantum Computing via Density-based Basis-set Correction.

Corresponding Author: Professor Jean-Philip Piquemal

Version 0:

Reviewer comments:

Reviewer #1

(Remarks to the Author)

Dear Editor,

I am writing to provide my review of the manuscript entitled "Shortcut to Chemically Accurate Quantum Computing via Density-based Basis-set Correction," submitted for consideration in Communications Chemistry. I appreciate the opportunity to review this work. The paper discusses a novel approach to quantum computing in the context of electronic structure calculations, which is critical for advancements in drug design and materials science. The paper appears to present significant advancements in the integration of quantum computing with quantum chemistry, demonstrating a novel methodology and achieving impressive results.

Below, I have outlined my comments and questions concerning the comparison and result analysis to assist in improving the manuscript.

Overall Assessment:

The authors introduce a density-based basis-set correction (DBBSC) method that integrates density-functional theory into quantum algorithms. This method utilizes dynamically generated basis sets tailored to specific systems and qubit budgets. The authors demonstrate the effectiveness of their approach using GPU-accelerated state-vector emulation with up to 32 qubits, achieving chemical accuracy for the ground-state energies of various small molecules.

Key Findings and Importance:

- **Results:** The authors tested their method on several small molecules and achieved highly precise results that would typically require much larger quantum computers. The method successfully converged the ground-state energies of He, Be, H₂, and LiH to chemical accuracy.
- **Impact:** The study significantly reduces computational resource requirements, making complex quantum-chemistry calculations feasible with fewer qubits. This advancement opens new possibilities for practical quantum-chemistry simulations, potentially leading to significant advancements in drug design and materials science by enabling more precise and efficient electronic structure calculations.

Minor Concerns and Questions:

- **Abstract:** The abstract effectively summarizes the study's objectives, methods, results, and conclusions. However, it would benefit from additional clarity regarding:

o **Objective Definition:** Could the authors elaborate on how their proposed method addresses existing gaps in the field?

o **Results Summary:** Could the authors include more quantitative details or key metrics to underscore the significance of their findings?

- **Comparison Table:** To enhance the clarity and completeness of the comparison presented in the table, I recommend adding a column detailing the quantum gate counts used in the ADAPT-VQE and other quantum algorithms evaluated. This addition would provide a more comprehensive understanding of the resource requirements and efficiency of each quantum algorithm, allowing for a fair comparison between different methods and better highlighting the trade-offs between algorithm complexity and performance.

"I recommend citing the paper by Burton, H. G. (2024) titled 'Accurate and gate-efficient quantum ansätze for electronic states without adaptive optimization' (Physical Review Research, 6(2), 023300). This paper provides valuable insights into optimizing qubit usage and improving gate efficiency in quantum computations. Citing this work would provide a useful benchmark for comparing qubit count and performance metrics with the ADAPT-VQE methodology discussed in your study. It could enhance the understanding of how different quantum algorithms perform in terms of qubit efficiency and overall computational effectiveness."

- **Suggestions and Future Directions:** While the study shows promising results, there are still challenges to address. The

method should be tested on a wider variety of chemical systems and with more qubits to ensure it can handle larger, more complex problems. Additionally, further work is needed to implement this method on actual quantum computers, not just in simulations. Overall, this study is a significant step forward in making quantum computing a practical tool for chemistry, but more research is needed to fully realize its potential.

Thank you for considering my feedback. I hope that the suggested revisions make a significant impact on the manuscript and contribute to the journal's goals.

Reviewer #2

(Remarks to the Author)

This manuscript addresses the question of how to approach the complete-basis-set (CBS) limit in quantum computation of electronic structure problems with limited resources (qubit count). The proposed strategy consists of integrating density-based basis-set correction (DBBSC) with general quantum algorithms, focusing on variational quantum ground state optimization algorithms that can already run on current Noisy Intermediate Scale Quantum hardware. The first ingredient of this integration is the addition of correction terms to any Hamiltonian's ground state energy approximation in a given finite basis set, adapting the DBBSC approach to the framework of quantum computing. These corrections account for two characteristic features of small basis sets: the non-convergence of the Hartree-Fock energy to the CBS limit, and the underestimation of short-range

electron correlations. The second ingredient of the proposed DBBSC integration is a novel atomic-orbital system-adapted basis sets (SABS) under basis-size budget with controllable accuracy. This alternative to the standard Dunning basis sets used in the literature of DBBSC is tailored for applications to quantum computing with limited qubit counts. The claim of the authors is to achieve energy corrected ground-state energies with quantum calculations in SABS on par with large double-zeta or even triple-zeta basis sets, while requiring much less quantum hardware resources.

The authors present convincing arguments that the combination of SABS, Hartree-Fock and density-based energy corrections provides a viable path to improve the accuracy of current quantum algorithms on near-term quantum devices while avoiding the large jump in qubit numbers required by standard basis sets. A few comments on these arguments:

- Recent works on transcorrelated methods [e.g. <https://doi.org/10.1103/PhysRevA.102.010601>] have also discussed how to approach the CBS limit in variational quantum computation. Including a comparison of this class of methods to the proposed strategy would help to better understand the positioning of this manuscript in the context of previous literature.

- An appealing feature of the approach is that the proposed correction terms can be computed without any overhead in the number of qubits for the quantum algorithm. The Hartree-Fock correction is evaluated a posteriori on a classical processor, and the density-based correction can either be computed a posteriori at the density field theory level, with a marginal classical computational cost, or self-consistently integrated in the hybrid classical-quantum optimization routine by adding a one-electron potential to the energy landscape. It would be interesting to mention to what extent the ADAPT-VQE calculations still satisfy a form of variational principle after having assumed a PBE density functional.

- While the authors do not propose any calculations on actual noisy quantum hardware, they provide results of classical emulation of quantum computing algorithm on graphics-processing-units (GPUs) for small molecules in increasingly large basis sets up to 30 qubits. Their converged quantum circuit optimization results are based on state-of-the-art ADAPT-VQE routines. These results allow them to separately quantify the energy improvement when adding the correction terms for any basis set and to compare of the qubit scaling and accuracy of the proposed system-adapted basis sets and of standard basis-sets. As an example, they achieve for N₂ in a 30-qubit basis set accuracies lying between double- and triple-zeta basis sets that would otherwise require around 100 qubits. By providing additional details on the circuit depths of the optimized ADAPT-VQE ansätze for the various basis-set size, one could also provide estimates of the how suitable these quantum circuits are for the current noisy intermediate-scale quantum circuits.

Overall, this manuscript offers a clear proposal for integrating DBBSC in near-term quantum computing applications. In addition, some of the novelties of this paper have applications beyond the field of near term quantum computing. For example the system-adapted basis-set could also find applications in classical methods for quantum chemistry, and the integration scheme for the basis-set energy correction would also be relevant for fault-tolerant quantum computation in small basis-set.

I would therefore recommend this work for publication in Communication Chemistry.

Reviewer #3

(Remarks to the Author)

I co-reviewed this manuscript with one of the reviewers who provided the listed reports. This is part of a Communications Chemistry initiative to facilitate training in peer review and to provide appropriate recognition for Early Career Researchers who co-review manuscripts.

Reviewer #4

(Remarks to the Author)

The article "Shortcut to Chemically Accurate Quantum Computing via Density-based Basis-set Correction" proposed a framework that utilizes the density-based basis-set correction (DBBSC) method to improve the accuracy of classical/quantum electronic-structure eigensolver methods. Specifically, the authors proposed two strategies.

Strategy one corrects the ground-state energy approximation produced by the applied eigensolver at once. The second

strategy is a self-consistent method that can iteratively update DBBSC (including HF energy discretization error). It can also be used with a system-adaptive basis set (SABS) that updates within the self-consistent iteration. The article provided proof-of-concept numerical demonstrations of computing small atoms and molecules' properties (ground-state energy, dissociation curve, and dipole moment) with the GPU-emulated ADAPT-VQE method.

Overall, the article is well-structured, and the authors presented their translational application of DBBSC in the context of quantum/classical simulation for chemical systems with an effective basis set compression technique.

However, I would like the authors to address my following comments and questions about some technical details of their research.

Comment 1:

In the paper, the author provides two corrections to the energy estimation. First is the basis-set correction density function based on the one-body reduced density (HF density). Second is the direct correction to the finite-basis error from the HF reference state. The author mentioned that the first correction only accounts for "short-range electron correction," which is unclear about its relationship to the error from the finite-basis HF energy. HF approximation can capture electron correlation contribution to the ground-state energy to a certain degree and, hence, can also benefit from the first correction in theory. So, "short-range electron correction" is not a sufficient reason to explicitly include a second correction for the HF energy term. Whether in the form of rigorous math proof or numerical simulations, the authors should show the significance of including the second correction beyond their simple explanation in the current manuscript.

Moreover, the authors used the HF energy with cc-pV5Z to approximate the CBS-limit HF energy. It's better to extrapolate the CBS limit based on multiple correlation-consistent basis set results, as they are less costly than the rest of the procedure. If the authors argue this is not necessary, they should at least provide data to show that the error of HF on cc-pV5Z is within the chemical accuracy compared to its actual CBS limit.

Comment 2:

For strategy two, the authors of the paper point out that it can be combined with actual quantum algorithms for ground-state estimation, where the (one-body reduced) density is obtained from the quantum solver. However, in practice, this is not trivial to implement. Specifically, when encoding the target molecular Hamiltonian to a qubit (spin) Hamiltonian on the quantum circuit, depending on the encoding techniques (first-quantization vs. second-quantization, local vs. nonlocal), the resulting many-qubit state may not preserve the structure of the original many-electron state, even though they share the same eigen spectrum. So, subsequently, the reduced density matrix's computation might also be inefficient. The author should at least provide their specific implementation of retrieving the density in their GPU emulator test case to demonstrate a realistic scenario of their techniques.

Comment 3:

In section 1-B, the authors mentioned related work of performing basis optimization to compensate for the qubit number limit from quantum devices (Ref 31, 32). However, the author claimed that these basis set optimizations cannot be performed on the fly, which does not seem to be supported by the content of references. The authors should remove such a claim unless they can further explain their reasons or provide quotations from the original papers.

Version 1:

Reviewer comments:

Reviewer #1

(Remarks to the Author)

Dear Editor

I have carefully reviewed the author's revisions, and I am pleased to confirm that all my concerns and suggestions have been thoroughly addressed. The manuscript now meets the necessary standards, and the changes have significantly improved the clarity and quality of the work. I am satisfied with the responses and revisions provided. Therefore, I am happy to recommend the acceptance of this revised version.

Reviewer #2

(Remarks to the Author)

Dear authors and dear editor,

The manuscript has been significantly improved following the remarks of the reviewers.

In particular, the authors have added a paragraph in the introduction and a table in the appendix to answer our questions on the comparison to trans-correlated methods and on the depth of the proposed quantum circuits.

We believe that this additional content helps understand the significance of this manuscript in the context of current literature and of current quantum hardware capabilities.

We therefore recommend the publication of the revised manuscript in Communications Chemistry.

Reviewer #3

(Remarks to the Author)

I co-reviewed this manuscript with one of the reviewers who provided the listed reports. This is part of a Communications Chemistry initiative to facilitate training in peer review and to provide appropriate recognition for Early Career Researchers who co-review manuscripts.

Reviewer #4

(Remarks to the Author)

I thank the authors for their detailed responses to my comments and suggestions. My main concerns for the manuscripts have been appropriately addressed, so I do not have further questions.

I. REVIEWER 1

A. Reviewer comments

1. Objective Definition: Could the authors elaborate on how their proposed method addresses existing gaps in the field?
2. Results Summary: Could the authors include more quantitative details or key metrics to underscore the significance of their findings?
3. Comparison Table: To enhance the clarity and completeness of the comparison presented in the table, II recommend adding a column detailing the quantum gate counts used in the ADAPT-VQE and other quantum algorithms evaluated. This addition would provide a more comprehensive understanding of the resource requirements and efficiency of each quantum algorithm, allowing for a fair comparison between different methods and better highlighting the trade-offs between algorithm complexity and performance.
4. I recommend citing the paper by Burton, H. G. (2024) titled 'Accurate and gate-efficient quantum ansätze for electronic states without adaptive optimization' (Physical Review Research, 6(2), 023300). This paper provides valuable insights into optimizing qubit usage and improving gate efficiency in quantum computations. Citing this work would provide a useful benchmark for comparing qubit count and performance metrics with the ADAPT-VQE methodology discussed in your study. It could enhance the understanding of how different quantum algorithms perform in terms of qubit efficiency and overall computational effectiveness.

B. Answers

We would like to thank the reviewer for his/her careful reading of our manuscript. We reply in the following paragraph to the remarks.

1. Nowadays the aim of designing quantum computing algorithms for quantum chemistry is linked to the necessity to provide useful applications on present NISQ and future FTQC quantum computers. In this context, the algorithms are build under some constraints: i) the need to have shallow circuits to reduce hardware qubit noise and, ii) an overall requirement over minimizing quantum resources such as the number of qubits. With the SABS and the density-based basis-set correction, we propose to reach improved accuracy on energies and dipole moments by using a computationally cheap classical approach. In fact, the adapted basis sets are obtained within seconds using small computational resources and allow us to reach ground-state energies with higher accuracy, while the DBBSC and Hartree-Fock corrections can be calculated with only the input data (the Hartree-Fock density). Please note that since the DBBSC itself uses usual density-functional theory, it does not increase the qubit count while offering improved electronic density thanks to its coupling with the quantum ansatz. The abstract has been rewritten to provide a better definition of our objective.
2. One way to answer this question is to consider the scaling of the number of operators with the number of qubits which has been defined in Ref. [1] as being $\mathcal{O}(N^4)$ with N the number of qubits. The number of CNOT gates in the qubit operator pool being $\propto n_{\text{ops}}^2$ (n_{ops} is the number of one-qubit operators). Then, the scaling of the number of CNOT gates with the number of qubits is N^8 . We think that the new table which has been added in Appendix (see Table IV) now provides useful key metrics to underscore the present discussion. We added a sentence in the text at the beginning of the Results section (page 7, left column): "We remind the reader that these calculations are to be interpreted in the framework of perfect (noiseless) logical qubits, and therefore the discussed errors can be only attributed to the method used. Table IV presents the quantum circuits that were used for all computations and provides the number of CNOT gates."
3. We thank the reviewer for this comment. We added a new table, see Table IV, in the Appendix at the end of the paper, that encompasses the number of CNOT gates.
4. The reviewer question deals with the problem of resource savings and the suggested paper is part of the strategies used on this purpose where the authors propose an alternative to adaptive ansatz. Here, the tiled unitary product states is shown to provide a faster convergence than standard fermionic ADAPT adaptive methods. We added the reference in the introduction (now reference 26). Note that following the demand of reviewer 2, the introduction is now presenting in detail the various alternatives (transcorrelated methods, alternative of ADAPT-VQE, etc..). Indeed, on the same topic, one could mention the Jastrow Ansatz (UCJ) (Ref. [2] for the LUCJ variant)

but also, all strategies based on distributed quantum computing (<https://doi.org/10.48550/arXiv.2408.05351>), “improved basis sets, exploitation of symmetries, or partitioning methods”, as listed in Ref. [3].

On the same line, we can add that the SABS and DBBSC strategies are also aimed at improving accuracy while saving quantum resources. However, they follow different routes and can be used with existing strategies:

- SABS are a new family of basis sets that can be built on-the-fly (i.e. in seconds) thanks to a pivoted Cholesky decomposition approach to provide more chemistry information (based on the Hartree-Fock density) than the equivalent standard basis sets with same number of basis functions. Therefore, it can be used to initialize the one-body and two body-integrals for any ansatz, including the tUPS one pointed by the reviewer.
- DBBSC presents the only requirement of using the reduced density matrix (rdm). Here we only use the 1-rdm, but approximations exist using the 2-rdm in the classical quantum chemistry literature. It could be easily applicable on top of a tUPS ansatz.

II. REVIEWERS 2 AND 3

A. Reviewers comments

1. Recent works on transcorrelated methods [e.g. <https://doi.org/10.1103/PhysRevA.102.010601>] have also discussed how to approach the CBS limit in variational quantum computation. Including a comparison of this class of methods to the proposed strategy would help to better understand the positioning of this manuscript in the context of previous literature.
2. An appealing feature of the approach is that the proposed correction terms can be computed without any overhead in the number of qubits for the quantum algorithm. The Hartree-Fock correction is evaluated a posteriori on a classical processor, and the density-based correction can either be computed a posteriori at the density field theory level, with a marginal classical computational cost, or self-consistently integrated in the hybrid classical-quantum optimization routine by adding a one-electron potential to the energy landscape. It would be interesting to mention to what extent the ADAPT-VQE calculations still satisfy a form of variational principle after having assumed a PBE density functional.
3. While the authors do not propose any calculations on actual noisy quantum hardware, they provide results of classical emulation of quantum computing algorithm on graphics-processing-units (GPUs) for small molecules in increasingly large basis sets up to 30 qubits. Their converged quantum circuit optimization results are based on state-of-the-art ADAPT-VQE routines. These results allow them to separately quantify the energy improvement when adding the correction terms for any basis set and to compare of the qubit scaling and accuracy of the proposed system-adapted basis sets and of standard basis-sets. As an example, they achieve for N₂ in a 30-qubit basis set accuracies lying between double- and triple-zeta basis sets that would otherwise require around 100 qubits. By providing additional details on the circuit depths of the optimized ADAPT-VQE ansätze for the various basis-set size, one could also provide estimates of the how suitable these quantum circuits are for the current noisy intermediate-scale quantum circuits.

B. Answers

We would like to thank the reviewers for their careful reading of our manuscript. We reply in the following paragraph to the remarks.

1. We thank to the reviewer for this remark. We reshaped our introduction and added a new paragraph dedicated to transcorrelated methods. The proposed reference is included.
2. The DBBSC and SABS strategies leverage the use of classical resources to reach a better accuracy while minimizing quantum resources. For the DBBSC, we act on the one-electron integrals, and thus on the second quantized Hamiltonian. For the SABS, we act on the atomic basis set used to build the molecular orbitals. Therefore, the proposed overall strategy can be used in parallel with any approach aiming to optimize the circuit cost, as the one introduced in the suggested reference. If SABS can be used for any quantum algorithm based on molecular orbitals, DBBSC variants might require some additional investigations to ensure the validity of the imagined process if they are not variational algorithms. With the self-consistent PBE correction (and without the non-self-consistent HF correction), Eq. (8) of our manuscript, the basis-set corrected energy is variationally defined since there is a minimization. However, due to the use of the PBE correction, the resulting energy cannot be guaranteed to be an upper bound of the exact energy. This is often a source of confusion for people because in usual wave-function methods, the variationality of the energy and the upper bound property are often synonymous. In a DFT context, we keep only the variationality of the energy but as soon as an approximate density functional is used the upper bound property is lost.
To make it clear to the reader, we added a sentence on page 4 (Section 1.A.2, before equation 8): "This leads to the following approximation to the ground-state energy that can be applied to any variational quantum ansatz"
3. We answered this remark by adding a new table (Table IV) in the Appendix present at the end of the paper (see discussion with reviewer 1, points 2 and 3).

III. REVIEWER 4

A. Reviewer comments

1. In the paper, the author provides two corrections to the energy estimation. First is the basis-set correction density function based on the one-body reduced density (HF density). Second is the direct correction to the finite-basis error from the HF reference state. The author mentioned that the first correction only accounts for "short-range electron correction," which is unclear about its relationship to the error from the finite-basis HF energy. HF approximation can capture electron correlation contribution to the ground-state energy to a certain degree and, hence, can also benefit from the first correction in theory. So, "short-range electron correction" is not a sufficient reason to explicitly include a second correction for the HF energy term. Whether in the form of rigorous math proof or numerical simulations, the authors should show the significance of including the second correction beyond their simple explanation in the current manuscript.
2. Moreover, the authors used the HF energy with cc-pV5Z to approximate the CBS-limit HF energy. It's better to extrapolate the CBS limit based on multiple correlation-consistent basis set results, as they are less costly than the rest of the procedure. If the authors argue this is not necessary, they should at least provide data to show that the error of HF on cc-PV5Z is within the chemical accuracy compared to its actual CBS limit.
3. For strategy two, the authors of the paper point out that it can be combined with actual quantum algorithms for ground-state estimation, where the (one-body reduced) density is obtained from the quantum solver. However, in practice, this is not trivial to implement. Specifically, when encoding the target molecular Hamiltonian to a qubit (spin) Hamiltonian on the quantum circuit, depending on the encoding techniques (first-quantization vs. second-quantization, local vs. nonlocal), the resulting many-qubit state may not preserve the structure of the original many-electron state, even though they share the same eigen spectrum. So, subsequently, the reduced density matrix's computation might also be inefficient. The author should at least provide their specific implementation of retrieving the density in their GPU emulator test case to demonstrate a realistic scenario of their techniques.
4. In section 1-B, the authors mentioned related work of performing basis optimization to compensate for the qubit number limit from quantum devices (Ref 31, 32). However, the author claimed that these basis set optimizations cannot be performed on the fly, which does not seem to be supported by the content of references. The authors should remove such a claim unless they can further explain their reasons or provide quotations from the original papers.

B. Answers

We would like to thank the reviewer for his/her careful reading of our manuscript. We reply in the following paragraph to the remarks.

1. First, we use the term correlation as the difference between the exact electronic system and its description using the Hartree-Fock method. In terms of energy, it translates as

$$E_{\text{correlation}} = E_{\text{exact}} - E_{\text{HF}}.$$

This is the most common definition in the quantum chemistry community and it is the one we use in this paper. To facilitate the reading of the article and avoid blurring the message, we have chosen not to mention the definitions of Fermi hole which describes the correlation between two electrons of the same spin in a Hartree-Fock wave function.

In principle, we could include all basis-set corrections (HF and correlation) in a single density functional. However, in practice, the PBE-based correction only takes care of the correlation part. This is due to the very construction of this functional which starts from the usual PBE correlation energy and modifies it at short-range (between two electrons) for adapting it at the amount of short-range correlation already present in the chosen basis set. Nothing in the construction of this PBE correction is made for correcting the basis-set error of the HF energy. Moreover, this HF basis-set error is not a short-range electron-electron effect (it is not due to the electron-electron cusp) so it would be probably very hard to find a good local density functional approximation for correcting it. Indeed, local density functional approximations are usually not very accurate for non-short-range electron-electron effects. For more details on the use of a Hartree-Fock basis-set correction in addition to

the PBE-basis basis set correction, see Ref. [4, 5].

We thank the reviewer for this comment and, in order to prevent any confusion for the readers, we added a sentence in Section I.A.1: “Let us emphasize again that the PBE-based basis-set correction mentioned above only takes care of the correlation part and therefore does not correct the basis-set error of the HF energy. Hence, there is no double counting between the two basis-set corrections, as shown in Refs. [51, 52].”

2. According to Ref. [6], it is better to use a large basis set than using an extrapolated scheme for the Hartree-Fock energy. In practice, regarding the target precision, we consider that the Hartree-Fock energy is converged if the difference between the cc-pV5Z and the cc-pV6Z energies is less than 1.6mHa (the chemical accuracy), which is the case for most of the systems. However, since the reviewer comment is relevant: we added the values of the cc-pV6Z Hartree-Fock energies in the supplementary information. Since the observed differences were very small, we kept the cc-pV5Z reference energies as it is. We have added a citation to Ref. [6] before Eq. (7) of the manuscript to justify our choice.
3. The one-particle density matrix elements are:

$$n_{pq,\alpha} = \langle \Psi | \hat{a}_{p,\alpha}^\dagger \hat{a}_{q,\alpha} | \Psi \rangle,$$

where α is the spin (we have the same expression for spin β), and p and q are molecular orbital indices. To compute the density matrix in quantum computing, one only needs a Jordan-Wigner mapping of the operator $\hat{a}_{p,\alpha}^\dagger \hat{a}_{q,\alpha}$ and to prepare the qubit states using the parameterized ADAPT-VQE ansatz. However, while increasing the number of qubits, we noticed computational costs issues starting around 20 qubits. Therefore, we implemented a sparse version of this workflow that will be detailed in a coming publication to introduce our Hyperion software. The matrix is then used to classically update the Hamiltonian and to classically compute the dipole moments. Thanks to the reviewer remark, we added a paragraph in the Methodology section (see text associated to Eq. 13) of the paper to clarify our implementation.

4. We agree with the reviewer that the references do not mention their computational costs and that we based our claim on our local tests of their software. Therefore we removed the claim. Please note that our pivoted Cholesky decomposition approach allows us to define basis sets in seconds. To the best of our knowledge, usual basis-set optimization schemes can take hours or even days to provide solutions. We made clear that our approach is fast in the conclusion: “An additional reduction of the required basis-set size is provided by our SABS approach. Beside being fast, as SABS can be generated within seconds, such a “black-box” pivoted-Cholesky strategy for the on-the fly generation of basis sets has been shown to be competitive and often superior to available standard choices.”

-
- [1] C. Feniou, M. Hassan, D. Traore, E. Giner, Y. Maday, and J.-P. Piquemal, “Overlap-ADAPT-VQE: Practical Quantum Chemistry on Quantum Computers via Overlap-Guided Compact Ansätze,” *Communications Physics*, vol. 6, 2023.
 - [2] M. Motta, K. J. Sung, K. B. Whaley, M. Head-Gordon, and J. Shee, “Bridging physical intuition and hardware efficiency for correlated electronic states: the local unitary cluster jastrow ansatz for electronic structure,” *Chemical Science*, vol. 14, no. 40, pp. 11213–11227, 2023.
 - [3] G. M. Jones and H.-A. Jacobsen, “Distributed quantum computing for chemical applications,” *arXiv preprint arXiv:2408.05351*, 2024.
 - [4] A. Hesselmann, E. Giner, P. Reinhardt, P. Knowles, H.-J. Werner, and J. Toulouse, “A density-fitting implementation of the density-based basis-set correction method,” *Journal of Computational Chemistry*, 2024.
 - [5] D. Mester and M. Kállay, “Basis set limit of CCSD(T) energies: Explicit correlation versus density-based basis-set correction,” *Journal of Chemical Theory and Computation*, vol. 19, no. 22, pp. 8210–8222, 2023.
 - [6] A. Halkier, T. Helgaker, P. Jørgensen, W. Klopper, and J. Olsen, “Basis-set convergence of the energy in molecular hartree–fock calculations,” *Chemical Physics Letters*, vol. 302, p. 437, 1999.